# The Impacts of Animal Health Service Providers on Antimicrobial Use Attitudes and Practices: An Examination of Poultry Layer Farmers in Ghana and Kenya

**DOI:** 10.3390/antibiotics9090554

**Published:** 2020-08-28

**Authors:** Kofi Afakye, Stella Kiambi, Eric Koka, Emmanuel Kabali, Alejandro Dorado-Garcia, Ann Amoah, Tabitha Kimani, Benjamin Adjei, Mark A Caudell

**Affiliations:** 1Food and Agriculture Organization of the United Nations, Accra 1628, Ghana; kofi.afakye@fao.org (K.A.); Ann.Amoah@fao.org (A.A.); Benjamin.Adjei@fao.org (B.A.); 2Food and Agriculture Organization of the United Nations, Nairobi 00100, Kenya; stella.kiambi@fao.org (S.K.); Tabitha.Kimani@fao.org (T.K.); 3Department of Sociology and Anthropology, University of Cape Coast, Cape Coast 5007, Ghana; ekoka@ucc.edu.gh; 4Food and Agriculture Organization of the United Nations, 00153 Rome, Italy; Emmanuel.Kabali@fao.org (E.K.); Alejandro.DoradoGarcia@fao.org (A.D.-G.)

**Keywords:** antimicrobial use, poultry farming, health seeking, Ghana, Kenya, antimicrobial resistance, antimicrobial stewardship

## Abstract

International organizations and governments have argued that animal health service providers can play a vital role in limiting antimicrobial resistance by promoting the prudent use of antimicrobials. However, there is little research on the impact of these service providers on prudent use at the farm level, especially in low- and middle-income countries where enforcement of prudent-use regulations is limited. Here, we use a mixed-methods approach to assess how animal health-seeking practices on layer farms in Ghana (*n* = 110) and Kenya (*n* = 76) impact self-reported antimicrobial usage, engagement in prudent administration and withdrawal practices and perceptions of antimicrobial resistance. In general, our results show that the frequency of health-seeking across a range of service providers (veterinarians, agrovets, and feed distributors) does not significantly correlate with prudent or non-prudent use practices or the levels of antimicrobials used. Instead, we find that patterns of antimicrobial use are linked to how much farmers invest in biosecurity (e.g., footbaths) and the following vaccination protocols. Our results emphasize that more research is required to understand the interactions between animal health service providers and farmers regarding antimicrobial use and antimicrobial resistance. Addressing these gaps will be crucial to inform antimicrobial stewardship training, curriculums and, guidelines whose ultimate purpose is to limit the selection and transmission of antimicrobial resistance.

## 1. Introduction

Animal health service providers (AHSP) can play a critical role in limiting the emergence and global spread of antimicrobial resistance (AMR). As the World Organization for Animal Health (OIE) states; “Being in contact with both animals and farmers, you are the frontline on the battlefront of antimicrobial resistance.” (emphasis added) [1]. Acknowledging this importance, global strategies to combat AMR, including those proposed by the OIE, Food and Agriculture Organization of the United Nations (FAO), and World Health Organization (WHO), all committed to supporting AHSPs in limiting the spread of AMR [1,2,3]. This support often takes the form of training and curriculums to motivate prudent antimicrobial use (AMU) because the use and misuse of antimicrobials is the primary selective force driving AMR [4,5]. AHSPs can promote prudent use of antimicrobial drugs by ensuring the correct diagnosis is made and the appropriate drugs are administered, and by advocating for other prudent practices, including the need to observe withdrawal periods from antimicrobials, proper drug disposal, and general biosecurity principles that may reduce disease pressures and thus demand for antimicrobials. In support, studies from high-income countries that track prescriptions and diseases treated provide evidence that AHSP involvement in animal health care is associated with the prudent use of antimicrobials [6]. In addition, veterinary stewardship programs such as the Yellow Card scheme in Denmark have been associated with decreased usage of antimicrobials [7,8] although the assessment of veterinary antimicrobial stewardship programs has generally been limited [9].

Within most low- and middle-income countries (LMICs), and particularly sub-Saharan Africa, the lack of prescription tracking systems means that national-level assessments of the impact of AHSPs and stewardship programs on prudent use over time are unfeasible. Instead, these relationships need to be ascertained from cross-sectional surveys. Most surveys provide evidence of the relationship between AHSPs and prudent use through assessing self-reported knowledge, attitudes, and practices (KAP) of AHSPs and those in-training [10,11]. For example, surveys of veterinarians and advanced veterinary students in Nigeria found that less than 25% had adequate knowledge of antibiotic stewardship strategies [10]. In a survey of practices, community animal health workers (CAHWs) in Ghana and Mozambique administered drugs at the correct dosage approximately 40% of the time while paraprofessionals in Uganda and Kenya were observed using antibiotics non-prudently [12]. Importantly, these findings are not confined to LMICs (see [13] for an example from Italy) but within LMICs weaker regulatory and enforcement contexts may mean that gaps in knowledge and practices have a greater impact on antimicrobial stewardship at the farm level.

To advance the understanding of the role of AHSPs in antimicrobial stewardship within LMICs, we used a mixed-methods approach to assess how animal health-seeking practices on layer farms in Ghana and Kenya impact self-reported antimicrobial usage, engagement in prudent administration and withdrawal practices and perceptions of AMR. We define AHSPs as individuals who have received formal training in veterinary sciences and/or receive compensation for services linked to animal health, including attendants at shops who sell veterinary medicines (agrovets). Our main objective was to assess how the frequency of health-seeking from government veterinarians, private veterinarians, community/extension officers, agrovets, and feed distributors was associated with self-reported AMU across an average month of production, adherence to withdrawal practices, and AMR-relevant knowledge and perceptions on the farm controlling for other potential factors that could impact these practices (i.e., disease burdens, farm economics, etc.). Both qualitative and quantitative results suggest that the frequency of seeking advice from AHSPs is largely unrelated to prudent practices and attitudes, and that monthly AMU was more strongly associated with biosecurity, disease pressures, and vaccination rates than health-seeking practices. We discuss potential reasons underlying the lack of association between health-seeking and AMU practices and perceptions. Building upon these reasons, we conclude by providing guidance on the future efforts required to ensure AHSPs can play a major role in limiting the emergence and spread of AMR.

## 2. Materials and Methods

### 2.1. Study Locations

Study locations in Ghana and Kenya were chosen given the prominence of layer production in the area and evidence from previous studies of the self-administration of antimicrobials as well as AMR in fecal samples and residues in eggs and water in drinkers [14,15,16]. The Kenyan KAP survey was conducted among 76 layer farmers in Gatundu North Subcounty, Kiambu County (see Figure 1). The economy of Kiambu County is dominated by agriculture with 75% of the population under a small-scale production system [14]. Kiambu County has the largest broiler and layer production systems in the country with Gatundu North Subcounty, where this study was conducted, having the highest number of layers. Farmers in Gatundu North are served by one government vet officer and one government animal health assistant. The Ghana KAP survey was carried out among 109 layer farmers in Dormaa Central Municipal of Bono Region, Ghana (see Figure 1). About 68.4% of households in the municipality are engaged in agriculture (21). It is the highest commercial poultry producing district in the country with about 3.1 million layers and 1 million broilers [17]. Livestock farmers in Dormaa district are served by government veterinary staff including one District Veterinary Officer (DVO) and five Veterinary Technical Officers (VTOs). All surveys took place between December 2018 and March 2019.

### 2.2. Survey Development

The study was designed by an interdisciplinary research team comprising animal health experts and social scientists from FAO, the Directorate of Veterinary Services, Directorate of Livestock Production, and Sub County animal healthcare workers in Kenya; and the Veterinary Services Directorate of Ministry of Food and Agriculture, and Regional and Municipal Veterinary Officers in Ghana. A mixed-methods approach combining qualitative and quantitative data collection methods was used to develop the KAP survey. In Kenya, this included conducting separate focus group discussions (FGDs) with male and female farmers (2), and key informant interviews (KIIs) with agrovets (2), government and private veterinary officers (2), and animal health assistants (1). In Ghana, qualitative methods included FGDs among two mixed-gender groups of farmers and veterinary officers in the municipality. Seven KIIs were also conducted among the following stakeholders: agrovets (2), feed millers (2), animal health assistants (1), and government veterinary staff (2). In both countries, FGDs and KIIs were concentrated around twelve major themes relating to AMU and AMR including farm management and economic practices, disease histories, and knowledge, attitudes, and practices relating to AMU and AMR, including governance, regulations, policies, and enforcement.

Thematic analysis of qualitative interviews was used to develop a KAP survey instrument of over 200 items that included a broad range of questions covering demographics, livelihood, health, and hygiene and biosecurity topics related to factors that could drive AMU and AMR. (See Appendix A for KAP questionnaire). KAP surveys were administered by a group of local enumerators (*n* = 4 in Ghana and *n* = 6 in Kenya) using tablets with the Kobo Collect^®^ application. Enumerators were taken through a comprehensive four-day training and piloting of the KAP questionnaire. The survey was administered in English, Kiswahili or Kikuyu (Kenya), or Twi (Ghana) with the respondent indicating what language they were most comfortable in using. Interviews lasted around one hour.

### 2.3. Sampling Procedure

In both countries, census records with production data were not available for the study areas (Gatundu North, Kenya and Dormaa, Ghana). In Kenya, enumerators conducted a door-to-door survey within local neighborhoods, generating a list of households from which a random selection of households was done and provided to enumerators. This census did not include all layer farmers in Gatundu North as some residents incorrectly believed that enumerators were a part of the county’s efforts to count the number of livestock for tariff calculation and therefore, refused to participate. In Ghana, a snowball sampling approach was used whereby veterinary officers were initially contacted to provide names of individuals currently engaged in layer production. Farmers that were listed during the snowball sampling were then consulted during the KAP survey to provide names of individuals who were currently engaged in layer production. This process was conducted until the data collection period ended (i.e., two weeks).

### 2.4. Ethical Approvals

Ethical approvals were obtained in each country. In Kenya, the study was approved by the AMREF Health Africa Ethics and Scientific Review Committee (AMREF-ESRC P551/2018). This research was also approved by the Institutional Animal Care and Use Committee of KALRO-Veterinary Science Research Institute, Muguga, upon compliance with all provision vetted under and coded: KALRO-VSRI/IACUC016/28092018. In Ghana, permission to conduct the study was approved by the Ministry of Health Ethical Review Board (ID No. 014/10/18).

In both countries an information sheet (IS) containing a detailed narrative of the study and its aims was provided to potential participants who could read and was read out to those who could not. Participants were informed of the research purposes including the benefits and risks of participation. The respondents were assured of their right to withdraw from study participation at any point, and necessary precautions were made to ensure and maintain confidentiality, anonymity, and voluntarism throughout the study. Written informed consent was sought and obtained from all study participants that could write. For those who could not write, a thumbprint signature was requested.

### 2.5. Analysis

A multivariate ordinal logistic model was used to assess associations between the number of times a farmer reported using antimicrobials during an average production month and health-seeking practices (see Table 1). Health seeking practices included in the model were variables representing the frequency of seeking advice/services across a range of AHSPs, including agrovets, feed distributors, community/extension livestock health officers, government veterinarians, and private veterinarians. Variables were recoded into binary values with 0 indicating that a provider was not sought for advice and treatment and 1 indicating the provider was sought for advice and treatment. Control variables included flock size (standardized), the number of diseases reported as common on the farm, the number of diseases the flock was reported to be vaccinated against, and a scale of biosecurity. The biosecurity variable is a linear scale that combines responses to questions on ownership of personal protective equipment (PPE) including gumboots, gloves, and overcoat, availability and type of footbath or tire bath, frequency of drinker and feeder cleaning, disinfection practices, and use of isolation chambers for sick birds. Good practices were coded as 1 and poor practices as 0 so that the biosecurity scale indicated the number of good practices the respondent followed. For the ordinal logistic model, results are presented as odds ratios (where >1 indicates increased odds and <1 decreased odds). Models were specified in Stata 16.1 [18].

To assess the relationship between health-seeking variables and knowledge, attitudes, and practices, Fisher’s exact tests were used. See Table 1 above for definitions of health-seeking variables and Table 2 below for knowledge, attitudes, and practices variables. Fisher’s exact tests were used to correlate these variables because some health-seeking variables and prudent practices/attitudes cross-tabulations had less than 5 observations. After running Fisher’s exact tests, post-hoc analyses were then performed to determine what associations between health-seeking frequency and knowledge, attitudes, and practices were driving significant associations as identified by the Fisher’s exact tests. These post-hoc comparisons were adjusted for multiple comparisons using the Benjamini–Hochberg false discovery rate [19]. Fisher’s exact tests and post-hoc analysis were conducted in RStudio [20]. All tables were created using the Stata program *asdoc* [21].

## 3. Results

### 3.1. Results: Descriptive

Compared to surveyed Ghanaian farmers, surveyed layer farmers in Kenya were older (average 50 years versus 36 years) and were more likely to be female (50% females versus 23% females). Ghanaian farmers kept the largest flocks with a median flock size of 4054 birds (Q_1_ = 2000, Q_3_ = 9000) while Kenya farms had a median of 700 birds (Q_1_ = 300, Q_3_ = 1150). Consistent with these farm size differences, Ghanaian farmers had more employees (≈4) compared to Kenya (≈1) and more poultry houses (≈11 versus ≈2 houses, respectively) (see Table 3). More Ghanaian farmers used a cage system (26%) but the deep litter system was favored in both Ghana (78%) and Kenya (98%). Kenyan farmers were more likely to keep multiple cohorts of animals with 88% of respondents saying they kept multiple cohorts compared to 44% of Ghanaians. In both Kenya and Ghana, most layer farmers also kept other animals, 95% and 71% respectively. Almost all Ghanaian farmers kept records (98%) while slightly more than half of Kenyan farmers were reported to keep records (59%).

Diseases reported to be common were similar across Kenyan and Ghanaian layer farmers except for chronic respiratory disease (CRD), which was listed by about 20% more Ghanaian farmers and Fowl Typhoid, which was listed by only 2% of Ghanaian farmers but 13% of Kenyan farmers (see Table 4). Coccidiosis was listed by around 60% of layers farmers in Kenya and Ghana while 30% to 40% listed Newcastle Disease and around 30% listed Infectious Bursal Disease (Gumboro). The least common diseases in both countries were Marek’s Disease (Ghana: 5%, Kenya: 4%) and Infectious Bronchitis (Ghana: 3%, Kenya: 8%).

Antimicrobial use, costs, and prudent practices also varied across countries (see Table 5). As expected, given the scale differences, Ghanaian farmers spent more and used more antimicrobials than their Kenyan counterparts. In terms of prudent AMU practices, differences existed in the use of antimicrobials as prevention, consuming products within withdrawal periods, and in securing prescriptions.

### 3.2. Results: Qualitative

Finding 1: In Kenya, agrovets are important sources of information for layer farmers while veterinarians are perceived as measures of “last resort”. 

In Kenya, discussions on health-seeking behavior revealed that farmers sought help mainly from the agrovets and rarely from veterinarians. Our FGDs among farmers supported this role as some referred to agrovets as “veterinarians”. Agrovets also mentioned that they were filling a void given the lack of government veterinary services in the area. A common refrain from the agrovets was “who else will do it?” Most agrovets had good relationships with government veterinarians with all believing that the lack of resources meant everyone needed to support animal health. In most cases, the farmer explained to the person at the agrovet the signs they were seeing among the flock and medicines were dispensed over the counter. Sometimes, the farmers carried a dead or sick bird to the agrovet where a postmortem was done to inform treatment options for the rest of the birds. One of the farmers said, “Sometimes, most of us farmers take fecal material or sick/ dead birds to the agrovets for diagnosis and buying of medicines”.

Veterinarians were said to be rarely contacted and only as the last resort as they were perceived to be doctors for large animals, mainly for cows and not for poultry. Indeed, the terminology they used for veterinarians was “daktari wa ng’ombe”, which is a Kiswahili word that literally translates to “doctor for cows”. In both FGDs, it emerged that even the private veterinarians who run ambulatory services focused only on the cows whereas the agrovets were said to be ready to help poultry farmers. One of the farmers, whom most FGD participants agreed with, said, “If you see a veterinary doctor in your area, there must be a rabies case in the village or a cow-related disease like anthrax”.

According to key informant interviews with AHSPs, most of them agreed that animal health information was usually attained first by peer-networks and those networks informed treatment and medicine purchasing patterns. Often, the farmers would take empty medicine containers that have been used by other farmers (who told them of its effectiveness) and insist on buying the same for their chickens. In some cases, the AHSPs would request to visit the farm in order to make a proper diagnosis but most farmers would decline (they would be asked to pay for services in the case of agrovets and sometimes fuel for motorbikes) and insist on purchasing the medicine. AHSPs agreed that the danger of relying on peer networks was that the farmer may end up either underdosing or overdosing the birds. A telling example provided was the use of Macrolan^®^, a product containing the antibiotic tylosin tartrate. A farmer relying on information from other farmers incorrectly administered 1 teaspoon/20 L of water as opposed to 1 tablespoon/20 L of water. An agrovet estimated that only one-third of poultry farmers in the area seek advice from either a veterinary professional or agrovets with the rest relying mostly on advice from peers and/or past experience.

Finding 2: Ghanaian layer farmers rely mostly on their own experiences to treat birds and believe they only require laboratory diagnostic services to verify diseases.

In Ghana, qualitative data on practices of seeking help from AHSPs revealed that most farmers did not rely much on agrovets nor veterinarians but rather on their own experiences and other experienced farmers (i.e., friends and neighbors) for acquiring animal health information. Indeed, narratives from the participants indicated that most have ample knowledge of antibiotics and mostly self-administer these products to their birds. However, because they lack access to laboratory facilities to diagnose the diseases affecting their birds, they give different kinds of antibiotics to them on a trial and error basis. Some of the antibiotics mentioned are Antibact ^®^ (Tylosin, Oxytetracycline, Neomycin), Doxin^®^ (Doxycycline and Tylosin), tylodoxin^®^ (Tylosin and Doxycycline) and Oxytetracycline.

Ghanaian farmers also voiced their frustration over the perceived inadequacy of professional veterinary care services. One farmer said, “If the veterinary staff had been doing their work well and then be checking on those who are not on the right path of farming and stop them irrespective of their money it will help prevent the spread of the disease (FGD, P10)”. This lack of service means that extra steps are taken, sometimes non-prudent, to ensure bird health. Recalling one of these practices, a farmer had this to say, “Sometimes when you go to the feed-mill and you see the number of drugs someone is coming to mix with the feed, it is like someone who is going to establish a drug-store. Some are even adding drugs meant for humans, without knowing its repercussion (FGD, P2)”. Commenting on the lack of laboratory services, another added that if they had facilities for proper diagnosis that they would act appropriately, saying “Not that we the poultry farmers don’t know the proper use of antibiotics. It is not the case that as soon as you identify a sick animal, you administer antibiotics. Buying the antibiotics is even a huge cost to us. We don’t have a choice because we don’t have a laboratory here for proper disease diagnosis. That is why we buy the antibiotics and use”.

In the case of Ghana, it appears that farmers do not believe they need health advice from agrovets or veterinarians but only results of diagnostic tests that a laboratory would provide. The implication of this is that most farmers based their diagnosis and treatment of sick birds on experience rather than a proper diagnosis in the laboratories. The main concern of farmers has to do with the non-availability of laboratory facilities for them to properly diagnose the diseases before administering drugs. The implication of this is that farmers will continue to engage in risky behaviors though some may be aware of the consequences since they have no option available to them.

### 3.3. Quantitative Analysis

#### 3.3.1. Impact of Animal Health Seeking Practices on Non-Prudent Practices and Knowledge and Perception

Fisher exact tests indicated that the frequency of seeking animal health advice across AHSPs was not significantly related to most prudent knowledge, attitudes, and practices associated to AMU and AMR. Table 6 provides results of measures of knowledge, attitudes, and practices that were significantly associated with the reported frequency of seeking animal health advice across provider types. Results within the table indicate post-hoc analyse’s comparing levels of health-seeking across KAP responses. For example, the row labeled “always gets a prescription” compares the number of respondents who said they “almost always” get a prescription across the different frequencies of seeking health advice from agrovets (i.e., never/rarely, sometimes, almost always). Frequencies that share a letter are not significantly different by Fisher’s exact tests. For example, in the row labeled “always gets a prescription”, the letter associated with “never/rarely” is “a”. As the letters associated with “sometimes” also included “a” (i.e., “ab”) the people who said they never/rarely sought the advice of agrovets and those who said sometimes sought advice did not significantly differ in their prescription seeking practices. However, the letter associated with frequency “almost always” in the “always gets a prescription row” is “b” so people who said they never/rarely sought the advice of agrovets and those who said “almost always” sought advice did significantly differ in their prescription seeking practices.

Although few trends emerged from the Fisher’s exact tests, there was evidence that animal health-seeking practices were related to having prescriptions for antimicrobials. Respondents who more frequently sought advice from agrovets, private veterinarians, and government veterinarians were more likely to report always having a prescription. Interestingly, those who more frequently sought advice from community/extension animal health workers were more likely to consume poultry products during withdrawal periods. Finally, those who more frequently sought advice from private veterinarians were less likely to report using antimicrobials to boost egg production and promote bird growth.

#### 3.3.2. Impact of Animal Health Seeking Practices on Antimicrobial Use

No AHSP seeking practices were significantly associated with levels of AMU on the farm, but AMU was related to the number of diseases reported on the farm as well as vaccinations and biosecurity practices (see Table 7). For every additional disease reported, the odds of using antimicrobials three to five times in a normal production month were 31% greater compared to the combined odds of other categories (never, one to two times) given other variables are held constant. Likewise, for every additional disease reported. the odds of the combined “three to five times” and “one to two times” were 31% higher than those reporting “never” given other variables are held constant. For every additional disease that was vaccinated against, the odds of using antimicrobials three to five times versus the combined odds of other categories (never, one to two times) decreased by 64%. Finally, for every additional biosecurity step taken, the odds of using antimicrobials more three to five times versus the combined odds of other categories (never, one to two times) decreased by 19%. As with diseases and vaccines, this effect is consistent when comparing across categories (the combined “three to five times” and “one to two times” versus never).

## 4. Discussion

The results of this study suggest that seeking out advice from AHSPs does not necessarily impact prudent or non-prudent use knowledge, attitudes, and practices in layer farmers in Ghana and Kenya. Except for evidence that AHSPs are associated with farmers having prescriptions, we did not find consistent and significant associations between self-reported prudent use and withdrawal practices and frequency of seeking advice across the five types of AHSPs included in this study (i.e., government veterinarians, private veterinarians, extension officers, feed distributors, and agrovet employees). These findings are somewhat consistent with knowledge, attitudes, and practices studies of AHSPs that document variable understandings of antimicrobial stewardship and impacts on prudent use, and so suggest that seeking advice from these providers may not improve prudent practices on the farm [10,23,24]. Combined with these earlier findings, this study emphasizes the need for antimicrobial stewardship training programs to ensure high-quality advice is provided by AHSPs. This is important as governments, NGOs, and international organizations have maintained that animal health service providers will play an important role in limiting the emergence and spread of AMR through promoting antimicrobial stewardship on the farm.

Where we did find associations between animal health-seeking practices and AMU-related knowledge, attitudes, and practices, they tended to be in unexpected directions. Those who more frequently sought advice from community animal health workers, for example, were more likely to engage in the non-prudent practice of consuming products within the withdrawal period. Importantly, the unexpected associations between health-seeking and knowledge, attitudes, and practices may mean that appropriate advice is not being shared with layer farmers in our study communities. Farmers may seek and receive prudent advice from AHSPs, but this advice may be disregarded due to economic interests. Ghanaian layer farmers, for example, knew that the selling of eggs during the withdrawal period was a sub-standard practice. However, there are currently no opportunities in the country for them to sell these products for alternative uses (e.g., soap production) to recover production costs, as occurs in some high-income countries. Moreover, Ghanaian and Kenyan farmers do not have access to insurance packages to compensate them in the event that they would have to dispose of the eggs during the withdrawal periods. As such, the economic costs of throwing away eggs during withdrawal periods may be too much to bear.

As demonstrated in the case of withdrawal, our results highlight that more research is needed on client-provider dynamics in veterinary care, particularly to understand the types of information exchanged between animal health professionals and livestock keepers. Post-hoc analysis of our data provides some insight into the information commonly exchanged during these interactions. For example, we asked farmers about their information exchanges with agrovets, given that most farmers (approx. 80%) reported buying antimicrobial drugs at agrovet shops. In Ghana, over 70% of farmers said they only tell the agrovet the name of the drug they need (i.e., they do not provide symptoms of sick birds) and are not provided instructions on use by the agrovet employee (dosage, treatment period, etc.). In contrast, over 90% of layer farmers in Kenya said they described the symptoms of their birds to the agrovet and were advised on the drugs to use and use instructions. Due to these differences in information exchange, seeking out agrovet services in Ghana may not have the same impact on prudent practices or perceptions as seeking these services in Kenya.

More broadly, the differences in animal health-seeking practices between Ghana and Kenya are also products of the differing farming scales of the studied communities. As economic investment and purchasing-power increases with larger farm sizes, so too does the motivation to fund proper animal care. In support, Ghana farmers were significantly more likely than Kenyan farmers to report seeking advice from trained veterinary professionals, including government, private, and extension veterinary staff. In addition, many Ghanaian households (i.e., 94%) reported that the costs associated with veterinary care were not a challenge. In contrast, small-scale Kenyan farmers may be less likely to invest heavily in animal healthcare. Indeed, Kenyan farmers said that the costs of veterinary care were a challenge (i.e., 72% of households). Our qualitative work in Kenya further suggests that farmers in smaller-scale layer systems, and possibly other poultry systems (broilers) may perceive veterinarians as those who only support large animal health and large-scale poultry operations.

### Future Work and Study Limitations

Research into the knowledge, attitudes, and practices of animal health professionals and on client-provider dynamics will be important in assessing the quality of animal health advice and in designing training programs, curriculums, and guidelines to promote antimicrobial stewardship. These steps are recognized as critical in AMR strategies of the FAO [2], OIE [1], and the WHO-led Global Action Plan for AMR, which includes the objective to “Establish antimicrobial resistance as a core component of professional education, training, certification, and development for the health and veterinary sectors and agricultural practice” [3]. Importantly, this training should not only be targeted at veterinarians but also to other actors, including agrovets and feed distributors, which studies show are common sources of advice on antimicrobials [22,25]. Further support for this training comes from a survey of agrovets in Africa and Asia, which found that knowledge of AMR did not translate into prudent use (e.g., reduced dispensing) but was actually linked to less prudent practices (e.g., use of critically important antibiotics) [24].

Our results also provide evidence-based insights into the potential content of AMU/AMR training, curriculums, and guidelines by identifying factors that promote AMU on poultry farms in Ghana and Kenya. For example, we found that AMU was negatively associated with vaccination rates and farm biosecurity levels. While these relationships are expected, training content should be developed emphasizing communication strategies to link vaccines, biosecurity, AMU, and AMR more explicitly, including the economic trade-offs of initial investments in biosecurity (e.g., footbaths) and vaccinations versus later costs for antimicrobials. To develop this targeted content, more research is needed on the specific biosecurity practices, vaccines, and diseases that are associated with AMU across different types of production systems and farm scales. In the current study, the small sample size meant that the number of variables that could be entered into models was limited and scales of vaccines, disease, and biosecurity had to be used. As such, we could not determine what specific practices, vaccines, or diseases drove the correlation with AMU or whether each exhibited similar effects. Likewise, to disentangle potential explanations behind the lack of association between health-seeking frequency and prudent AMU practices, larger-scale studies are needed to permit statistical regression techniques that can control for confounding factors (e.g., the impact of wealth on health-seeking practices) as well as assessments of the quality of advice received from AHSPs, which were not possible in the current study. More research on the knowledge, attitudes, and practices of common sources of animal health advice will allow a better understanding of whether the frequency of utilization will translate to behavioral change on the farm.

In addition, to link health-seeking and farm dynamics with AMU and AMR, better measures of AMU are needed to assess the impact of AHSPs on antimicrobial stewardship. Our measure of use, the self-reported number of times antimicrobials are used in an average production month, may have been affected by recall biases, as has been shown in other health-related studies [26]. Studies are needed that incorporate methods such as repeated self-report measures, passive surveillance methods, including the collection of used sachets and bottles in waste buckets, and sales data at local agrovet shops. Triangulation across these methods will help to assess the reliability and validity of quantitative AMU data. These more robust methods will also be necessary to identify the drivers of AMU, to parse out prudent from non-prudent use, and to better examine the relationship between AMU practices and the seeking of professional advice.

Finally, while stewardship training will be needed to improve the quality of advice from AHSPs, the success of these efforts will also depend heavily on parallel efforts to increase access to AHSPs in LMICs. Across the last 30–40 years, privatization of the animal healthcare sector within these countries, which was meant to improve the quality, efficiency, and coverage of services, has largely not compensated for service reductions in the public sector, [27,28]. According to the OIE, most LMICs do not have adequate veterinary service coverage [29,30]. Caudell et al. [22], for example, report that the ratio of veterinarians to livestock in five African countries (Ghana, Kenya, Tanzania, Zambia, and Zimbabwe) is about 20 times lower compared to European countries and the United States. These access issues are clearly observed in patterns of AMU and related practices on farms in LMICs, including the administration of drugs without prescriptions [16,25,31,32,33,34,35,36,37,38]. As such, the success of calls for additional training and prudent use guidelines by the FAO, OIE, and WHO must also be accompanied by strategies to increase access to AHSPs.

## 5. Conclusions

AHSPs will play an important role in impacting prudent antimicrobial use and thus reducing the emergence and spread of antimicrobial resistance globally. However, there is limited research attempting to understand the impacts of AHSPs on prudent use practices on the farm, especially in LMICs. Using ethnographic data from layer farmers in Ghana and Kenya, we showed that AHSP seeking frequency, to a large extent, was not associated with prudent use practices. Our results highlight that more research is required to understand the interactions between AHSPs and farmers, especially the quality of animal health advice receive.. Filling the gaps in these interactions will inform antimicrobial stewardship training, curriculums, and guidelines whose ultimate purpose is to limit the spread and transmission of AMR.

## Figures and Tables

**Figure 1 antibiotics-09-00554-f001:**
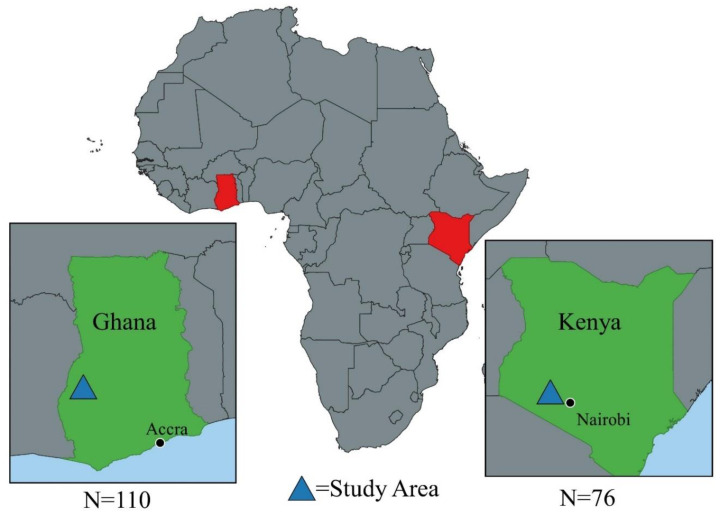
Map of the study areas Kenya and Ghana. See map legend for a description of map markers. Maps were created using QGIS. Base layers for the map were downloaded from © OpenStreetMap contributors http://www.vdsgeo.com/osm-data.aspx and licensed under Creative Commons Attribution—ShareAlike 2.0. The map was created using QGIS, version 3.14.

**Table 1 antibiotics-09-00554-t001:** Variables included in the multivariate ordinal logistic model. The variable column corresponds to the variable name in models. The level column indicates the coding of the variable (e.g., 0 = never for AM USE variable). For variables with levels of “continuous” please see Table 3 for ranges. All data are self-reported by survey respondents.

Variable and Role in Model	Definition	Levels
Practices		
AM USE -Outcome	The number of times antimicrobials were reported to be used in an average production month.	Frequency of use 0 = Never 1= 1–2 times 2 = 3–5 times
Agrovet Advice-Predictor	Whether an agrovet was sought for advice	0 = Advice not sought 1 = Advice Sought
Feed Distributor Advice-Predictor	Whether a feed distributor was sought for advice	0 = Advice not sought 1 = Advice Sought
Community/Extension Advice-Predictor	Whether a community animal health worker was sought for advice	0 = Advice not sought 1 = Advice Sought
Govt. Veterinarian Advice-Predictor	Whether a government veterinarian was sought for advice	0 = Advice not sought 1 = Advice Sought
Private Veterinarian Advice-Predictor	Whether a private veterinarian was sought for advice	0 = Advice not sought 1 = Advice Sought
Farm Size Standardized-Control	The number of layers on the farm standardized	Continuous
Number of Diseases Impacting Flock-Control	The number of diseases a farmer reported as “common” on the farm	Continuous
Total Number of Diseases Vaccinated against-Control	The number of diseases a flock was vaccinated against	Continuous
Biosecurity Scale-Control	A linear scale indicating the level of biosecurity on the farm. See text above for additional information	Continuous

**Table 2 antibiotics-09-00554-t002:** Knowledge, attitudes, and practices variables included in the Fisher exact test included in models. For health-seeking variables included in the model, see Table 1) AM stands for antimicrobials.

Variable	Definition	Levels
Practices		
Promoters	Respondent reported using antimicrobials to promote faster or larger growth	1 = Yes 0 = No
Boost eggs	Respondent reported using antimicrobials to boost egg production	1 = Yes 0 = No
Prevention	Respondent reported using antimicrobials to prevent animal getting diseases in the future	1 = Yes 0 = No
Group_treat	Respondents reported using the strategy of treating all birds if a few became sick	1= Yes 0 = No
Stop treat	Respondent reported stopping the recommended treatment period early if animal health improved	1 = Yes 0 = No
Prescrip	Respondent reported almost always getting a prescription before buying antimicrobials	1 = Yes 0 = No
Am docs	Respondent reported giving day old chicks antibiotics upon arrival to the farm	1 = Yes 0 = No
Consume	Respondent reported that products within the withdrawal period (eggs and meat) were consumed at home	1 = Yes 0 = No
Share	Respondent reported that products within the withdrawal period (eggs and meat) were shared with friends and family outside the home	1 = Yes 0 = No
Sell	Respondent reported that products within the withdrawal period (eggs and meat) were sold	1 = Yes 0 = No
Knowledge/perception		
Explain ams	Respondent explained the antimicrobials killed disease	1 = Yes 0 = No
Explain amr	Respondent explained antimicrobial resistance occurred when disease/germs could not be killed by drugs	1 = Yes 0 = No
Amr impact	How much a respondent believed that AMR would impact their future livelihood. (The question was only asked to respondents who could explain AMR (*n* = 56)).	0 = No Impact 1 = A little Impact 2 = A large Impact

**Table 3 antibiotics-09-00554-t003:** Summary statistics across Ghana and Kenya farms. Bolded variables indicate inclusion in models.

	Ghana *n* = 110				Kenya *n* = 76			
	Mean	SD	Min	Max	Mean	SD	Min	Max
Age	36.67	13.02	18.00	64.00	49.34	14.95	23.00	80.00
Gender (1 = Female)	0.23	0.42	-	-	0.53	0.50	-	-
Household Size	5.28	2.89	2.00	22.00	4.80	2.98	0.00	20.00
Employees	4.23	3.25	0.00	15.00	0.97	1.07	0.00	4.00
Years Keeping Layers	9.35	7.30	0.10	33.00	10.88	8.39	0.25	38.00
Flock Size	9037	14,795	10.00	10,000	1079	2292	90.00	19,000
Cage System (1 = Yes)	0.26	0.44	0.00	1.00	0.04	0.20	0.00	1.00
Deep Litter (1 = Yes)	0.77	0.42	0.00	1.00	0.99	0.11	0.00	1.00
Layer Houses	11.34	15.78	1.00	127.00	2.39	1.68	1.00	11.00
Keep Multiple Cohorts	0.44	0.50	0.00	1.00	0.88	0.33	0.00	1.00
Other Animals	0.71	0.46	0.00	1.00	0.95	0.22	0.00	1.00
Keep Records	0.98	0.13	0.00	1.00	0.59	0.49	0.00	1.00
**Farm Size (std)**	0.27	1.22	−0.48	7.78	−0.39	0.19	−0.47	1.09
**Diseases Reported**	2.85	1.69	0.00	9.00	2.64	1.84	0.00	7.00
**Vaccines Reported**	2.58	1.14	0.00	5.00	3.93	0.50	2.00	5.00
**Biosecurity Steps**	8.85	2.15	5.00	12.00	9.71	1.67	3.00	12.00

**Table 4 antibiotics-09-00554-t004:** Layer diseases reported. Data first reported in Caudell et al. 2020 [22]. Ghana *n* = 110. Kenya, *n* = 76. All percentages are rounded up to whole numbers.

	Ghana		Kenya	
	% Reporting	*n*	% Reporting	*n*
Coccidiosis	63	69	64	49
Infectious Coryza	21	23	14	11
Chronic Respiratory Disease	85	94	63	48
Fowl Pox	14	15	14	11
Fowl Typhoid	2	2	13	10
Gumboro	31	34	34	26
Helminthiasis (Worms)	18	20	14	11
Infectious Bronchitis	3	3	8	6
Marek’s Disease	5	6	4	3
Newcastle Diseases	43	47	34	26
My Birds Never Get Diseases	2	2	3	2
I Do Not Know Any Diseases	3	3	8	6

**Table 5 antibiotics-09-00554-t005:** Summary statistics on AM-use, non-prudent, withdrawal, and knowledge. All numbers are the percentage of farmers responding “yes” for non-prudent and withdrawal practices or correctly for knowledge/perception questions. All percentages are rounded up to whole numbers. AM is antimicrobial.

	Ghana *n* = 110		Kenya *n* = 76	
	% of HH	*n*	% of HH	*n*
AM USE: NEVER	9	10	63	48
AM USE: 1 to 2 times	77	85	33	25
AM Use: 3 to 5 times	14	15	4	3
PROMOTE	13	14	11	8
BOOST EGGS	35	39	11	8
PREVENTION	53	58	37	28
GROUP TREAT	82	90	92	70
STOP TREAT	30	33	12	9
NO PRESCRIP	35	39	66	50
AM DOCs	69	76	74	56
CONSUME	75	83	39	30
SHARE	33	36	18	14
SOLD	93	102	87	66
EXPLAIN AMs	84	92	83	63
EXPLAIN AMR	58	64	53	40
AMR IMPACT Not Worried (=0)	9	10	4	3
AMR IMPACT A Little Worried (=1)	25	28	3	2
AMR IMPACT Very Worried (=2)	24	26	46	35

**Table 6 antibiotics-09-00554-t006:** Significant differences in prudent knowledge, attitudes, and practices across health-seeking frequencies. The first column reports the animal health professional (AHSP) followed by the frequency of seeking health advice from that AHSP. The second column reports the practice or attitude included in the Fisher’s exact test. The third column reports the percentage of people who reported seeking a certain frequency who responded “yes” to the practice or attitude. For example, in the second row, 11.83% of respondents who reported going to an agrovet “never/rarely” said they always get a prescription (i.e., “yes”). The fourth column represents whether significant differences exist across the health-seeking frequencies on a practice/attitude. Frequencies (i.e., never/rarely, sometimes, almost always) that share a letter are not significantly different by Fisher’s exact test *p*-values adjusted by the Benjamini–Hochberg false discovery rate for multiple comparisons [19]. See text for additional explanation and interpretation.

AHSP: Seeking Frequency	Practice/Attitude	% Responding “Yes”	Letter
Agrovet: never/rarely	Always get prescription	11.83	a
Agrovet: sometimes	Always get prescription	10.75	ab
Agrovet: almost always	Always get prescription	24.73	b
Extension: never/rarely	Consume withdrawal products	27.96	a
Extension: sometimes	Consume withdrawal products	17.20	b
Extension: almost always	Consume withdrawal products	15.59	b
Extension: never/rarely	Can explain AMR	29.35	a
Extension: sometimes	Can explain AMR	11.96	a
Extension: almost always	Can explain AMR	15.22	b
Feed company: never/rarely	Share withdrawal products	17.20	a
Feed company: sometimes	Share withdrawal products	9.14	b
Feed company: almost always	Share withdrawal products	0.54	a
Govt. veterinarian: never/rarely	Always get prescription	26.34	ab
Govt. veterinarian: sometimes	Always get prescription	3.76	a
Govt. veterinarian: almost always	Always get prescription	17.20	b
Priv. veterinarian: never/rarely	Give AMs to boost egg production	20.43	a
Priv. veterinarian: sometimes	Give AMs to boost egg production	3.76	a
Priv. veterinarian: almost always	Give AMs to boost egg production	1.08	b
Priv. veterinarian: never/rarely	Always get prescription	27.96	a
Priv. veterinarian: sometimes	Always get prescription	5.91	ab
Priv. veterinarian: almost always	Always get prescription	13.44	b
Priv. veterinarian: never/rarely	Use AM as growth promoters	8.06	ab
Priv. veterinarian: sometimes	Use AM as growth promoters	3.23	a
Priv. veterinarian: almost always	Use AM as growth promoters	0.54	b

**Table 7 antibiotics-09-00554-t007:** Monthly antimicrobial usage during the production cycle. An ordinal logistic model with an outcome variable of antimicrobial use (AMU) across a typical month as reported by respondents. AMU is categorized into 0 = never, 1 = 1 to 2 times, 3 = 3 to 5 times. 95% confidence intervals (CIs) are presented below odds ratio estimates. See Table 2 for variable definitions.

Variables	AMU Odds Ratio (95% CI)
Agrovet Advice	0.68 (0.34–1.35)
Feed Distributor Advice	1.65 (0.78–3.51)
Community/Extension Advice	0.63 (0.29–1.40)
Govt. Veterinarian Advice	1.86 (0.86–4.01)
Private Veterinarian Advice	1.08 (0.54–2.17)
Farm Size Standardized	1.13 (0.83–1.55)
Number of Diseases Impacting Flock	1.31 *** (1.09–1.58)
Total Number of Diseases Vaccinated Against	0.46 *** (0.33–0.63)
Biosecurity Scale	0.81 ** (0.69–0.96)
Observations	186

*** *p* < 0.01, ** *p* < 0.05.

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
