# Peer review of "The Impacts of Animal Health Service Providers on Antimicrobial Use Attitudes and Practices: An Examination of Poultry Layer Farmers in Ghana and Kenya"

_antibiotics, 2020, doi:10.3390/antibiotics9090554_

Round 1

Reviewer 1 Report

In the submitted manuscript, authors examined the impact of animal health service providers on antimicrobial use attitudes and practices with a focus on examination of layer farmers in Ghana and Kenya. Health providers definitely play a critical role in mitigating the spread of antimicrobial resistance, but limited research on the impact of such service providers on antibiotic prudent use in livestock production, particularly in low-and middle-income countries. This work sought to fill such knowledge gap. Data were collected and analyzed in a proper way, and the interpretation of results was scientifically sound.

While this study indicates that health service providers (the frequency of health seeking) does not correlate with prudent or non-prudent use practices, it provides useful data into this filed. As the authors stated, of course, larger studies were required to fully evaluated factors including health provider services on controlling overall antimicrobial resistance. I only have some minor comments.

  1. Line 109: layer farmers? Is this supposed to be 110? You have listed 110 in tables/figures.
  2. Line 297: typo here. Get rid of question mark.

Author Response

We thank the reviewer for their time in reviewing the manuscript. We have attached our responses. 

Reviewer 2 Report

To the authors: I was very interested to review this paper because of my interest in “One Health” and the high concern for antimicrobial resistance around the world. My initial comment is with the title of the paper. Although complete, the addition of “poultry layers” would clarify it.

In the abstract, there are several abbreviations used that are not defined  until later in the text. Initial use should be spelled out followed by abbreviation use. In line 29-30, “…patterns of antimicrobial…vaccination.” Is an awkward phrase that I am not sure of the meaning in the sentence.

In the introduction, line49 – “prudent use (of what) by ensuring”. Line 61 “longitudinal assessments” needs to be defined. Line 70 “…then farmers” ?. Much of the introduction is very difficulty to follow your thought process or directions. Please rewrite to clarify meaning  and English phrasing.

In Materialr, Line 98 “AMR in eggs and water” is this well water, ground water – runoff? Line 126 “FGDs” have not been defined.

Analysis Section – This is a very difficult section to discover the pertinent material from. I was able to decipher the Table 1 with some time and effort but I’m afraid that most readers will not take the time to work through it.

In the qualitative results section, the description tends to be very long and detailed, with the main points getting lost within the text. Perhaps the use of “bullet-points” at the end of the section would help to reinforce the qualitative findings.

The quantitative results are extremely difficult to parse out. The description of results from Table 5 don’t seem to fit what I am seeing and I would advise a complete re-do of Table 5 to make it somewhat user-friendly or interpretable. The sentence from lines 344-347 needs to be rewritten – I’m not sure of the intended meaning. Table 6 is also difficult to interpret and needs to be re-done. I don’t have a suggestion for a chart or graph form that would work with this model.

I’m not sure that the Discussion can be fairly evaluated because of the English phrasing problems and the need for a total re-write of this section. Very confusing in numerous places.

I was able to get most if the information from the paper, but the effort required will not be exerted by most readers. I think that the material is valuable on the differences in antimicrobial usage in these 2 countries and the attitudes of the poultry producer toward the animal health providers. I would encourage you to have an editor rewrite the paper for clarity and then resubmit.

Author Response

We thank Reviewer 2 for their time in reviewing the manuscript. Please find our response attached. 

Reviewer 3 Report

The manuscript concerns a problem with the level of knowledge and practices of usage of antimicrobial agents in poultry farm in two countries in Africa. It is clear and well written. The data are important because the most common mistakes are elicited with the study. The direction of work on order to improve the practices in poultry farms is also outlined.

Abstract: Page 2/29, Line 34: "spread" can be replaced by "selection"

In my view, the direct opinion of a single farmers should be rewritten. Please, do not cite the opinions, try to explain the problem and discuss the mistakes (see for example page 16/29, lines 309-312, lines 314-316 ....)

Table 3, fifth line: "02" to be changed to "2"

Author Response

We thank Reviewer 3 for their time in reviewing our manuscript. Please see our response attached. 

Round 2

Reviewer 2 Report

I want to congratulate the authors for the improvements in the paper from the original submission. The corrections and additional clarifications in the text make a tremendous difference in the clarity of the material. I appreciated the highlighting of a majority of the changes. I feel that the additions of the "Findings Header" in the Qualitative Results adds a great deal to this section. I feel that the Quantitative Results section is still somewhat difficult to interpret. The description of the significance column is helpful for Table 6 but the layout of the table does not lend itself for ease of interpretation. I assume that the "practice/attitude" statement only applies to the AHSP that is bolded above. I don't know if moving the AHSP to the left margin and dividing each of these sections with a line would help to clarify this table. I'm concerned that most readers will not take the time to understand this table. I could follow all of the other tables with no problems but Table 6 still needs some work. 

Table 5: AMR IMPACT A little worried - figures need to be aligned.

Line 374:"antimicrobial use on the farm, but it was related..."

Line 388: Move (AMU) to after antimicrobial use in line 387.

Lines 415: "Importantly, the unexpected association between health..."

Line 416: "may not mean that inappropriate advice..." Double negative "may mean that appropriate advice.."

Line 429: "exchanged"

Line 440: Space between Kenya and are.

Line 447: "Kenyan farmers..."

Line 469:"...drive AMU in poultry farms..."

Line 475-476:Are you inferring a direct effect in "...specific biosecurity practices, vaccines, and diseases that drive AMU...?

Line 489: "...measures of antimicrobial use..."

Once again, I appreciate your efforts in the rewrite. If possible I would really like to see some way to clarify Table 6.

Good Job!

Author Response

We again thank Reviewer 2 for their thoughtful consideration and constructive suggestions on our manuscript. We believe their suggestions have greatly improved the clarity of the manuscript. Please see our attached reply for our point-by-point response.  
